# Variational Multi-scale Representation for Estimating Uncertainty in 3D Gaussian Splatting

**Ruiqi Li, Yiu-ming Cheung***
Department of Computer Science, Hong Kong Baptist University
{csrqli, ymc}@comp.hkbu.edu.hk

## Abstract

Recently, 3D Gaussian Splatting (3DGS) has become popular in reconstructing dense 3D representations of appearance and geometry. However, the learning pipeline in 3DGS inherently lacks the ability to quantify uncertainty, which is an important factor in applications like robotics mapping and navigation. In this paper, we propose an uncertainty estimation method built upon the Bayesian inference framework. Specifically, we propose a method to build variational multi-scale 3D Gaussians, where we leverage explicit scale information in 3DGS parameters to construct diversified parameter space samples. We develop an offset table technique to draw local multi-scale samples efficiently by offsetting selected attributes and sharing other base attributes. Then, the offset table is learned by variational inference with multi-scale prior. The learned offset posterior can quantify the uncertainty of each individual Gaussian component, and be used in the forward pass to infer the predictive uncertainty. Extensive experimental results on various benchmark datasets show that the proposed method provides well-aligned calibration performance on estimated uncertainty and better rendering quality compared with the previous methods that enable uncertainty quantification with view synthesis. Besides, by leveraging the model parameter uncertainty estimated by our method, we can remove noisy Gaussians automatically, thereby obtaining a high-fidelity part of the reconstructed scene, which is of great help in improving the visual quality. [1]

## 1  Introduction

The radiance field methods [1] for view synthesis have received increasing attention in the past few years due to their capability of achieving photorealistic results. Among them, the recently proposed 3D Gaussian Splatting (3DGS) algorithm [2] has pushed the boundary of real-time view synthesis with prominent quality and efficiency. However, a major functionality deficiency of 3DGS is that it cannot provide uncertainty information regarding the reconstructed model and predictions. Such uncertainty information would be useful in removing the noisy components in the model and providing confidence maps to assess the quality view synthesis results, which is important in applications such as autonomous driving simulation and robotics navigation.

Previous works in view synthesis made attempts to quantify the uncertainty in Neural Radiance Field (NeRF) models via ensemble, variational inference or Laplace's approximation methods [3, 4, 5, 6]. Although these works demonstrate the ability to quantify the uncertainty with NeRF models, they cannot be directly applied to 3DGS models, due to the intrinsic difference between the implicit NeRF representation and explicit 3DGS. Furthermore, some prior works deal with the uncertainty in learning models with methods such as Monte-Carlo dropout [7], Deep Ensemble [8] and Subnetwork

---

*Corresponding author is Yiu-ming Cheung (ymc@comp.hkbu.edu.hk).
[1]Code is available at https://github.com/csrqli/variational-3dgs.

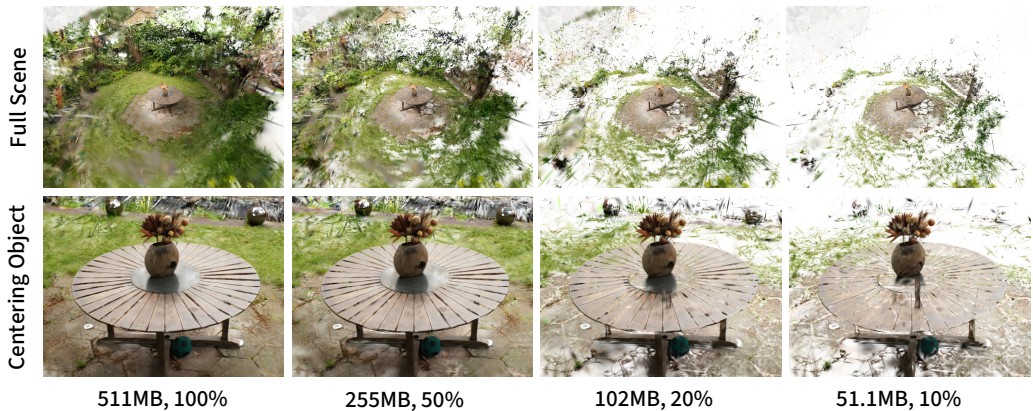

**Full Scene**

**Centering Object**

511MB, 100%        255MB, 50%        102MB, 20%        51.1MB, 10%

Figure 1: The results of cleaning up an unbounded scene reconstructed with 3DGS using our uncertainty estimation. We remove the Gaussians with large parameter uncertainty, the majority of which are under-reconstructed background. The desk at the center of the scene remains complete even after removing 90% of the Gaussians.

[9]. A major characteristic of these methods is that they can be seen as the approximation of Bayesian Inference, which estimates the distribution of posterior $p(\theta|\mathcal{D})$ and prediction $p(y|\theta, x)$, instead of point estimation.

The methods mentioned above can be seen as inferring the posterior distribution using model space samples. As mentioned in previous works [8, 10, 11], the key objectives for good model space samples are diversity and efficiency. To improve the quality of uncertainty estimation by increasing the diversity, the generated model space samples should explore the potential cases of true posterior as much as possible, which was not achieved in previous work on NeRF uncertainty estimation. Furthermore, although naive methods like ensemble provide good approximations, they require large storage and the computational cost which increase linearly with the number of samples. Thus, on top of diversity, efficiency is another important aspect we focus on when estimating the uncertainty for 3DGS, which means that we should use as few samples of parameters as possible.

In this paper, we design an uncertainty estimation method for 3DGS by constructing diversified parameter space samples efficiently. Inspired by the Level-of-Detail (LoD) technique used in computer graphics [12, 13] and multi-scale representation widely used in computer vision [14, 15], we propose to investigate the potential of scale information in fitting the scene representation with diversified samples. Specifically, we design a multi-scale variational inference method for 3DGS, which increases the diversity of parameter samples by enforcing them to model local spatial areas with multiple scales. To reduce the number of extra parameters, we design a mechanism that spawns finer multi-scale Gaussians from the base Gaussian, which are heavily involved in the rendering process. Instead of creating actual new Gaussians, we maintain an offset table parameterizing only a subset of attributes and share the remainder with base Gaussian. This can further reduce the number of parameter samples by maintaining only the attributes that contribute to our multi-scale representation.

Extensive experiments demonstrate the remarkable performance of our method on uncertainty estimation without affecting the rendering quality. We also show that our method provides both accurate posterior and predictive uncertainty estimation. The posterior uncertainty can be utilized directly to remove noisy Gaussians, thanks to the explicit benefit of the 3DGS method. The predictive uncertainty can serve as a confidence map to interpret the synthesized novel views.

The main contributions of our work are summarized as follows:

- We propose a multi-scale variational inference framework for uncertainty-aware view synthesis with the 3D Gaussian Splatting algorithm.
- We develop a spawning strategy to create multi-scale representations for 3DGS, and increase the sample diversity and inference efficiency by maintaining an offset table and sharing parameters.

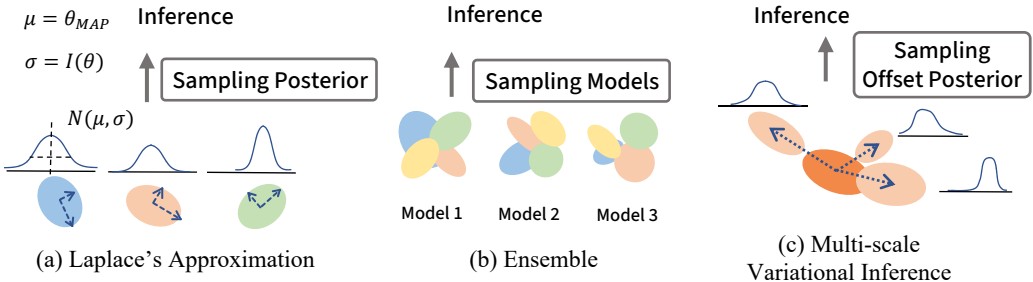

Figure 2: The comparison between our multi-scale variational inference and other methods. (a) Laplace's Approximation fits posterior with normal distribution where the mean equals maximum a posteriori solution $\theta_{MAP}$ and precision equals fisher information $I(\theta)$. (b) The ensemble method learns multiple models simultaneously to form the model space samples. (c) Our method builds a multi-scale representation of the scene, where inference is done by sampling the offset distribution and forming finer Gaussians.

- We evaluate the accuracy of uncertainty estimation on various benchmarks, and demonstrate the application of the parameter uncertainty in cleaning up noisy components in the scene.

## 2 Related Work

### 2.1 Uncertainty Quantification

Providing uncertainty together with predictions of learning models is of great use such as interpreting the model output [16, 17] or providing guidance in active data collection [18, 19]. This goal can be achieved by Bayesian learning, which characterizes the predictive distribution and posterior distribution of model parameters with theoretical foundations [20]. One can use Hamiltonian Monte Carlo sampling for Bayesian learning in neural networks [21], which guarantees asymptotic performance. Previous works proposed other approximation methods such as Laplace's approximation [22]. These methods consider the correlations between parameters, and are known for not requiring further assumption while reducing computations.

More recent works borrow from regularization techniques such as the dropout method as an approximated Bayesian inference, such as Monte-Carlo DropOut (MCDO) [7], Concrete Dropout [23] and Variational Dropout [24]. Another line of work builds model ensembles from various subsets of data [8, 16], hyperparameters [25] or multiple subnetworks [9, 26] to infer the uncertainty. Some methods also introduce network modulation to form a model ensemble, among which BatchEnsemble [27] learns multiple low-rank weight matrices. Turkoglu et al. [28] introduced a set of linear modulation parameters. Other works focus on extra network components, such as Variational AutoEncoders (VAEs) methods [29] or auxiliary network [30].

The commonality of the above methods is that they introduce randomness into the model and draw model space samples. Furthermore, the importance of the diversity of model space samples in approximated Bayesian learning is demonstrated in [11, 31]. Unlike the implicit characteristics of neural networks, the parameters of 3DGS have physical meanings that we can handle explicitly.

For algorithms in the multi-view geometry in computer vision, the uncertainty of camera pose can be estimated from the covariance matrix of the transformation matrix using Monte Carlo methods [32]. In the Simultaneous Localization and Mapping (SLAM) system, the uncertainty can be estimated from Kalman filter [33]. Our method can be aggregated to a dense SLAM system to provide uncertainty information for optimizing camera pose together with the map.

### 2.2 View Synthesis and Radiance Field

Recently, the radiance field has become popular in 3D vision, which achieves great success in generating photo-realistic images of novel view directions from a set of calibrated images. Early research

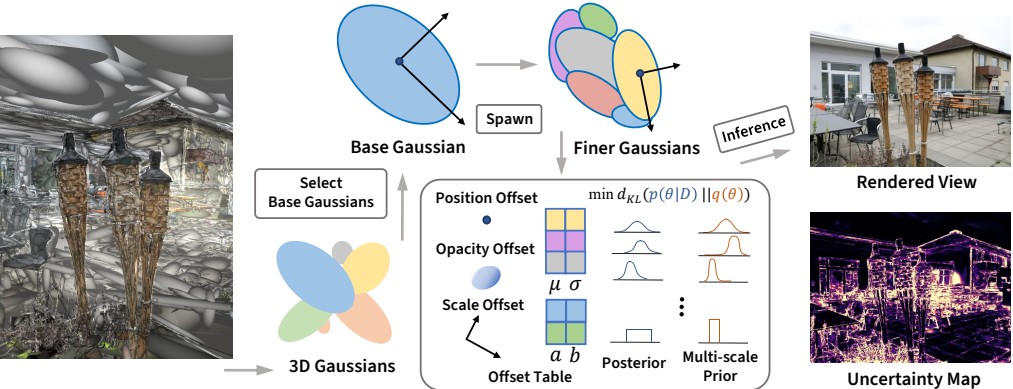

Figure 3: The pipeline of our variational multi-scale representation. We spawn base Gaussians, which are the major components in the scene, into multi-scale finer Gaussians. We learn an offset table to perform the spawn operation by offsetting a subset of attributes. The offset table is learned with variational inference with multi-scale prior. The predictive and parameter uncertainty can be inferred from the variational parameters stored in the table.

developed light field [34] and multiplane images based [35] methods for this task. Building upon previous research on implicit representations [36, 37, 38, 39], NeRF and its variants achieve a successful framework combining volume rendering and positional encoding to synthesize photorealistic novel views [1, 40, 41]. The recently proposed 3DGS [2] explicitly represents the 3D scene with points associated with the Gaussian function. Some works explore potential solutions to the anti-aliasing rendering problem, for example, Turki et al. [42] developed a multi-resolution feature grid, and Yan et al. [43] designed a multi-scale Gaussian with level-of-detail. In our work, we propose a variational multi-scale representation for 3DGS by spreading only local deviations to form multi-scale samples of model parameters.

Some previous works focus on uncertainty estimation for NeRF models. Stochastic NeRF [4] addresses the uncertainty estimation in NeRF models via standard variational inference technique. NeRF Ensembles [6] leverages neural network ensemble technique to quantify the predictive uncertainty. CF-NeRF [5] models the predictive distribution via learning a conditional normalizing flow. ActiveRMAP [44] focuses on the active reconstruction task, and models the predictive uncertainty via entropy of ray density distribution. NeurAR and ActiveNeRF [45, 19] adopt the neural network to output uncertainty values for each pixel and learning with Negative Log-Likelihood loss. ProbNeRF [46] designs a learned variational autoencoder that can generate 3D models from 2D images and uses the Monte Carlo method at inference to provide predictive uncertainty. Bayes' Ray [3] performs a post-hoc Laplace's approximation for NeRF to quantify the uncertainty via applying perturbations to spatial points. Compared to the NeRF models, the uncertainty estimation of 3DGS is less discussed. FisherRF [47] designs an uncertainty quantification method for 3DGS and applies it to active learning, which adopts Laplace's approximation to compute the uncertainty from Fisher Information. However, the approximation of posterior uncertainty needs extensive computations of gradient aggregating over the whole training views. In our work, we leverage variational inference to approximate the posterior distribution of parameters.

## 3 Proposed Method

### 3.1 Preliminaries: Uncertainty Quantification for 3DGS

The goal of the view synthesis problem is to generate images from any viewpoint given a set of calibrated input images. NeRF [1] proposes to solve this problem by representing the geometry and appearance of the scene using a learned radiance field $f(\mathbf{x}, \mathbf{d}) \rightarrow (\mathbf{c}, \sigma)$, where $\mathbf{x}$ and $\mathbf{d}$ represent a spatial point and view direction, $\mathbf{c}$ and $\sigma$ represent the corresponding color and opacity of that point viewing from $\mathbf{d}$. Based on this radiance field representation, 3DGS algorithm [2] further proposes

that the values in the field can be explicitly stored in a set of ellipsoids parameterized by the Gaussian function. The radiance value of each point in the scene $\mathbf{x}$ is queried from its adjacent ellipsoid from the Gaussian function:

$$\mathcal{G}(\mathbf{p}) = \exp\left(-\frac{1}{2}\left(\mathbf{x} - \mathbf{p}\right)^{\top} \Sigma^{-1} \left(\mathbf{x} - \mathbf{p}\right)\right),\tag{1}$$

where $\mathbf{p}$ and $\Sigma$ are the center and covariance matrix of the ellipsoid. The covariance is further decomposed to rotation $R$ and scale $S$ by $\Sigma = RSS^T R^T$. We referred to each ellipsoid as Gaussian $\mathcal{G}$. In the rendering process, each Gaussian component is projected to the image space and transformed into its 2D projections. After that, they are accumulated via alpha blending to form pixel values $\mathbf{c}$ in the rendered image:

$$\mathbf{c} = \sum_{n=1}^{N} \mathbf{c}_n \alpha_n \mathcal{G}_n(\mathbf{x}) \prod_{j=1}^{n-1} \left(1 - \alpha_j \mathcal{G}_j(\mathbf{x})\right).\tag{2}$$

Note that the rendering process is deterministic. For each Gaussian $\mathcal{G}$, its learnable parameters $\boldsymbol{\theta}$ are composed of position, color, density $\alpha$, and covariance: $\boldsymbol{\theta} = \{\mathbf{p}, \mathbf{c}, \alpha, S, R\}$. The scale $S$ and rotation $R$ are learned respectively and form the covariance matrix $\Sigma$ in rendering.

Standard 3DGS algorithm applies a non-Bayesian approach to train the model with $\mathcal{L}_1$ loss function, which can be seen as Maximum Likelihood Estimation (MLE) with the error following Laplace distribution [20]. However, this only performs point estimation which lacks the ability to quantify predictive uncertainty and provide confidence information.

Previous methods in Bayesian learning provide tools like variational inference and ensemble methods for estimating the predictive uncertainty of models. For example, with ensemble methods, we can train multiple 3DGS models with different data subsets, random seeds or hyperparameters, and compute the variance of their output as the predictive uncertainty. With variational inference methods, the model posterior $p(\boldsymbol{\theta}|\mathcal{D})$ is approximated by a tractable variational distribution $q(\boldsymbol{\theta})$, where $\mathcal{D}$ is the training data. Then, the likelihood of pixel color $p(\mathbf{c}|x, \boldsymbol{\theta})$ at pixel $x$ together with the discrepancy between $p(\boldsymbol{\theta}|\mathcal{D})$ and $q(\boldsymbol{\theta})$ is optimized.

### 3.2 Local Multi-scale 3D Gaussian

As discussed in previous works [8, 10, 11], the effectiveness of approximation methods in Bayesian inference highly depends on the diversity of model parameter space samples. With the explicit attributes of 3DGS parameters, we can manipulate the diversity of parameter samples and explore extensively the model parameter space. With more formations of the model representing the scene explored during learning, we can achieve better approximations in inferring the parameter posterior distribution. In the following, we will introduce a practical approach to building a representation with such diversification ability.

**Local Multi-scale Representation.**   Different from NeRF-based scene representation where the model parameters are the neural network parameters with no explicit meaning, 3D Gaussian models the local area of the spatial scene with attributes describing the geometry and appearance. Therefore, we can perform heterogeneous operations for Gaussian attributes by diversifying the scale of the local Gaussian to increase the performance in approximations. In our local multi-scale 3D Gaussian, we propose to learn scene representations with multiple Gaussian scales to represent a local spatial area. Specifically, we first select Gaussians that contribute more to the representation of the scene, and draw new multi-scale samples that are attached to these Gaussians. The parameters of the new Gaussians are shared or learned with variational inference. Through this strategy, we are able to control the scale variance and achieve diversified model space samples $\boldsymbol{\theta}$ in our uncertainty estimation.

**Spawn from Base Gaussians.**   The pipeline for spawning local multi-scale 3D Gaussians is illustrated in Figure 3. For every fixed step in training, we perform a spawn operation for Gaussians. Specifically, we would like to select the Gaussians with larger scales and more contributions to the

scene representation. We refer to them as *base Gaussians*, which is found by selecting Gaussians whose scale and opacity are above certain thresholds in the meantime. These base Gaussians are of greater contributions in the scene representation, which can be replaced by a set of Gaussians with various scales alternatively. Thresholding the gradient magnitude filters out trivial Gaussians such as those located in the distant background where they can hardly be seen.

**Learn Finer Gaussians via Offset Table.** After that, we spawn multi-scale finer Gaussians locally on top of base Gaussians. Instead of creating actual new Gaussians, to reduce the computational cost, we create a table $\phi$ to store the parameters of offset distribution for $K$ finer Gaussians. We find that some of the attributes of finer Gaussians can be shared as the same as base Gaussians with losing representation ability. We select position $\mathbf{p}$, scale $S$ and opacity $\alpha$ as the attributes maintained in the learned offset table, and the color $\mathbf{c}$ and rotation $R$ are shared by the base Gaussian associated. At inference, we apply the sampled offset to the base Gaussian to get the finer Gaussian. For example, the new position $\mathbf{p}^*$ and scale $S^*$ after offset are:

$$\mathbf{p}^* = \mathbf{p} + \chi_{\mathbf{p}}; \qquad S^* = S + \chi_S, \tag{3}$$

where $\chi_{\mathbf{p}}$ and $\chi_S$ represent position and scale offset values sampled from the offset distribution parameterized by $\tilde{\phi}$. There are total $K$ entries in the offset table $\phi$ while only $M$ of them are selected randomly with equal probability and averaged to get the final offset distribution parameters $\tilde{\phi} = \frac{1}{M} \sum_m^M \phi(i_m)$, where $i$ represents the index for the selected entries. By doing so, we create a subset of finer Gaussians with multiple scales and select a random mixture of their attribute distribution parameters each time. Therefore, we build a random scale alternative to the original large and significant base Gaussians. For the base Gaussians, the densification, splitting, and cloning operations are performed identically to the convention in [2], and these operations are also performed for the offset table. We will introduce how to learn the offset table with variance in scale using variational inference in the following.

## 3.3 Infer the Posterior of the Offset Table

The offset table to learn contains entries for $K$ finer Gaussians, and the inference process introduces randomness in selecting the $M$ entries that form the final offset distribution parameters. After obtaining these parameters, we perform variational inference for the offset distribution that enforces the finer Gaussains to be multi-scaled by assigning the prior distribution for the offset. We can infer the uncertainty from the distribution of $\boldsymbol{\theta}^*$, the parameter after applying the offset. Specifically, to infer the posterior and learn a multi-scale representation, we let $q(\chi)$ be the variational distribution for offset to approximate the true offset posterior distribution $p(\chi|\mathcal{D})$ and minimize the Kullback–Leibler divergence between them:

$$d_{KL}\left[q(\chi)\|p(\chi \mid \mathcal{D})\right] := \int_{\chi} q(\chi) \log \frac{q(\chi)}{p(\chi \mid \mathcal{D})} \tag{4}$$

$$= -\mathbb{E}_{\chi \sim q(\chi)}\left[\log p(\chi, \mathcal{D}) - \log q(\chi)\right] + \log p(\mathcal{D}), \tag{5}$$

where $p(\chi)$ is the prior distribution for offset $\chi$. For position offset $\chi_{\mathbf{p}}$ and scale offset $\chi_S$, we assume normal distribution and uniform distribution respectively as the prior. Particularly, to learn multi-scale samples, we assume the offset prior distribution with the following parameters:

$$q(\chi_S) \sim U(-S_{base} + (1 - 1/K)S_{base}, 0); \quad q(\chi_{\mathbf{p}}) \sim \mathcal{N}(0, \delta^2), \tag{6}$$

where $S_{base}$ is the scale of base Gaussian that the finer Gaussian attached, $\delta^2$ is the prior variance for the position offset. The range of scale after offset for a base Gaussian would be $[(1 - 1/K)S_{base}, S_{base}]$, which means that the scale after offset would vary with the associated base Gaussian. The lower bound of the prior is $(1 - 1/K)$ times the base Gaussian scale. This design choice ensures that the offset applied to the base Gaussian generates finer Gaussian with a larger scale range as the number of spawned Gaussians decreases, which means that larger diversity in scale is applied when there are fewer spawned Gaussians. Additionally, we choose zero mean value for the prior of position offset to enforce that the position lies around the base Gaussian, and the prior of

**Algorithm 1** The pseudo-code of the training process of our uncertainty-aware 3DGS.

---

**Input**: Images and corresponding camera poses
**Parameter**: Maximum training step $T$; Spawn interval $t$; Threshold $\tau$
**Output**: Trained scene representation with parameter $\boldsymbol{\theta}$; offset table $\phi$

1: **while** step $< T$ **do**
2:     **if** step % $t == 0$ **then**
3:         Select $\mathcal{G}_{base} = \{\mathcal{G}_n | \sum ||\nabla\boldsymbol{\theta}|| > \tau_{\boldsymbol{\theta}}, ||S_n|| > \tau_S, \alpha > \tau_\alpha\}$
4:         Spawn $\mathcal{G}_{base}$, create offset table $\phi = \{\phi_S, \ \phi_{\mathbf{p}}, \ \phi_\alpha\}$
5:         Assign prior $p(\chi)$ for offsets
6:     **end if**
7:     Sample offset $\chi$, render image $\mathbf{c}$ and compute image loss $\mathcal{L}_1, \mathcal{L}_{SSIM}$
8:     Compute KL divergence $\mathcal{L}_{KL} = d_{KL}(p(\chi|\mathcal{D})||q(\chi))$
9:     Optimize $\boldsymbol{\theta}, \phi$ with total loss $\mathcal{L} = \mathcal{L}_1 + \mathcal{L}_{SSIM} + \mathcal{L}_{KL}$
10: **end while**

---

opacity offset is given in the Appendix B. We learn the offset table $\phi$ and with the reparameterization trick [24]. The total loss function is $\mathcal{L}_{total} = \mathcal{L}_1(\hat{\mathbf{c}}, \mathbf{c}) + \mathcal{L}_{SSIM}(\hat{\mathbf{c}}, \mathbf{c}) + d_{KL}[q(\chi)||p(\chi)]$, where $\hat{\mathbf{c}}$ is the ground truth of color and $\mathcal{L}_{SSIM}$ is structural similarity index measure loss introduced in [2]. At inference, predictive distribution is inferred by marginalizing over the offset distribution:

$$p(\mathbf{c} \mid x, \mathcal{D}) = \mathop{\mathbb{E}}_{\chi \sim p(\chi|\mathcal{D})}[p(\mathbf{c} \mid x, \chi)] = \int p(\mathbf{c} \mid x, \chi)p(\chi \mid \mathcal{D})\mathrm{d}\chi, \tag{7}$$

where $p(\mathbf{c} \mid x, \mathcal{D})$ is the color prediction for pixel $x$. The pseudo-code of our proposer algorithm is shown in Algorithm 1.

## 4 Experiments

We evaluate our uncertainty-aware view synthesis technique on multiple real-world scenes. We compare the quantitative metrics for predictive uncertainty by estimating its correlation with the prediction error. Furthermore, we also validate the quality of our posterior uncertainty by removing Gaussians using different uncertainty threshold levels, which is quite practical for cleaning up and removing noisy regions of the scene, as known as floaters [48]. We will first introduce the datasets, metrics and baseline methods we used for evaluation, then the implementation details of our multi-scale variational inference algorithm. Finally, we will present quantitative results on uncertainty estimation, view synthesis and qualitative results on floater removal in the following.

### 4.1 Experimental Details

**Datasets.** We use three datasets for evaluation: **i) LF dataset** [49] contains in total 8 indoor and outdoor scenes, each containing over 100 images from $360°$ view. Following the same setting with CF-NeRF [5], we use images from selected scenes `torch`, `basket`, `africa`, `statue` for evaluation. **ii) LLFF dataset** [34] contains 8 forward-facing and outdoor scenes, each containing 20 to 62 images where the camera positions are arranged in a grid pattern. **iii) Mip-NeRF 360 dataset** [50] contains 6 outdoor scenes, in each scene, more than 200 images are captured in $360°$ view. The scenes in this dataset are unbounded and contain detailed regions like grass fields, which is challenging for reconstruction. We use this dataset to demonstrate the noisy Gaussian removal ability of our method.

**Evaluation Metrics.** For evaluating uncertainty estimation quantitatively, we first use the Area Under Sparsification Error (AUSE) with Mean Absolute Error (MAE) error. This metric evaluated the correlation between estimated uncertainty and true MAE error in each rendered view. It reorders the pixels according to the estimated uncertainty and calculates the difference between the reordered MAE array and the original ones. We normalize the error when calculating the difference. Secondly, we use the Negative Log-Likelihood (NLL) as a metric, which measures the likelihood of ground truth in the predictive distribution. This metric can evaluate both uncertainty and image quality at the same time. For image quality evaluation, we use Peak Signal-to-Noise Ratio (PSNR) to estimate

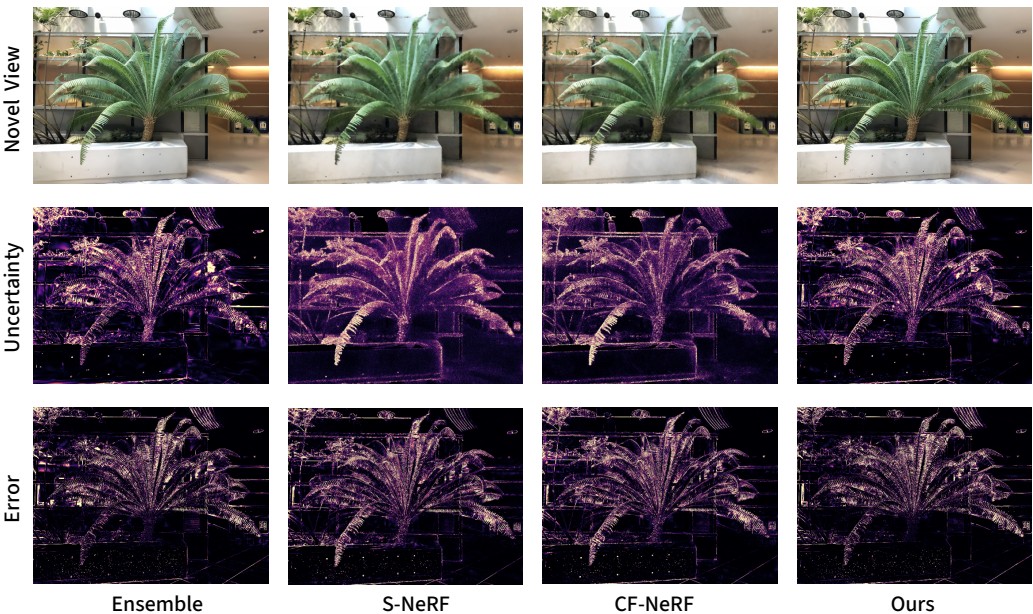

Figure 4: The visualization of predicted uncertainty map of novel view renderings. Our method demonstrates the best alignment of the uncertainty map with the error map.

the noise level, Structural Similarity Index Measure (SSIM) to measure the structural distortion and Learned Perceptual Image Patch Similarity (LPIPS) to measure the perceptual quality.

**Baseline Methods.** **i) CF-NeRF** [5] enables the uncertainty estimation of NeRF models by modeling latent variables and learning a conditional normalizing flows model. **ii) S-NeRF** [4] applies Bayesian learning to the NeRF model, and is able to quantify both depth and color uncertainty. **iii) Bayes' Ray** [3] constructs a spatial uncertainty field that can add perturbations to the position input of the radiance field and performs Laplace's approximation. **iv) Ensemble GS.** In this method, we train 10 3DGS models with different subsets of initialization points from Structure from motion (SfM) [51]. We also use different random seeds for each model. The variance of the predictions between all models is regarded as the predictive uncertainty.

**Implementation Details.** We use an AdamW optimizer to update the learnable parameters of our variational multi-scale representation. The learning rate for 3DGS attributes is the same as the original algorithm [2]. We choose to spawn $K = 10$ finer Gaussians in the offset table to perform our multi-scale variational inference. The learning rate of the offset table is 0.1 times the learning rate of each attribute. The experiments are performed on a single NVIDIA A100 GPU.

### 4.2 Uncertainty Estimation Quality Evaluation

We first evaluate the uncertainty of depth on the LF dataset. The calibration between uncertainty and depth error is quantified by AUSE reported in Table 1. On average, our method achieves the best performance. The performance of the ensemble method approaches our method while being better than other methods. However, our method requires much less computational cost compared to the model ensemble. In the basket scene, our method largely improves the AUSE metric by 0.9 compared to the second-best ensemble method.

Furthermore, we evaluate the uncertainty quality of rendered images on the LF and LLFF datasets. In Table 2, we report the AUSE and NLL metrics for rendered images. The NLL is estimated by the multivariate kernel density estimator used in [5]. On the LF dataset, our method shows the best AUSE, and outperforms S-NeRF in both metrics. In terms of NLL, CF-NeRF shows comparable results in aligning the predictive distribution with the ground truth. A plausible reason is that CF-NeRF models

Table 1: The depth uncertainty estimation performance on the LF dataset, quantified by the AUSE with MAE error.

| LF Dataset | africa | basket | statue | torch | Average |
|---|---|---|---|---|---|
| CF-NeRF | 0.35 | 0.31 | 0.46 | 0.97 | 0.52 |
| S-NeRF | 0.66 | 0.38 | 0.67 | 0.74 | 0.61 |
| Bayes' Ray | 0.27 | 0.28 | 0.17 | 0.22 | 0.23 |
| Ensemble GS ($\times 10$) | 0.16 | 0.22 | 0.17 | 0.26 | 0.20 |
| **Ours** | 0.19 | 0.13 | 0.21 | 0.23 | 0.19 |

Table 2: The performance of novel view rendering and uncertainty estimation on rendered images within the LF and LLFF dataset.

| | Method | Synthesized View Quality | | | Uncertainty Quality | |
|---|---|---|---|---|---|---|
| | | PSNR ↑ | SSIM ↑ | LPIPS ↓ | AUSE ↓ | NLL ↓ |
| LF Dataset | CF-NeRF | 24.32 | 0.835 | 0.202 | 0.49 | -0.37 |
| | S-NeRF | 20.21 | 0.761 | 0.248 | 0.62 | 1.32 |
| | Ensemble GS ($\times 10$) | 27.64 | 0.902 | 0.088 | 0.29 | -0.34 |
| | **Ours** | 27.39 | 0.914 | 0.101 | 0.26 | -0.30 |
| LLFF Dataset | CF-NeRF | 21.74 | 0.782 | 0.190 | 0.48 | 0.58 |
| | S-NeRF | 20.10 | 0.744 | 0.221 | 0.59 | 0.91 |
| | Ensemble GS ($\times 10$) | 24.54 | 0.810 | 0.157 | 0.30 | 0.26 |
| | **Ours** | 23.97 | 0.806 | 0.172 | 0.32 | 0.23 |

radiance distribution which helps improve performance. For the results on the LLFF dataset, our method demonstrates the best performance in terms of the NLL metric. We also surpass methods other than ensemble in terms of the AUSE metric. Bayes' Ray models spatial perturbation; therefore, only depth uncertainty quality is rendered and evaluated for this method. Our method does not offset color attributes directly to maintain efficiency, while still being able to model the uncertainty of predictive color by offsetting other attributes. We also provide visualizations of the predicted uncertainty map shown in Figure 4. Our method aligns well with the prediction error, even in detailed regions with complex scene content and overlapping object parts.

## 4.3 Rendering Quality Evaluation

Apart from the quality of uncertainty, we also compare the quality of synthesized novel views on both LF and LLFF datasets. The results on both datasets demonstrate the superior image quality of our method. Our method rivals the performance of the ensemble method in rendered image quality, without the need to learn an extensive number of additional parameters for Spherical Harmonics (SH) coefficients that store radiance information. In the LF dataset, our method shows the best quality in terms of SSIM.

We provide rendered images of the `fern` scene in Figure 4. Our method reconstructs the scene with clear details. The error pixels of the rendered image are located mainly in the region near the edge of the objects in the scene. A possible reason is that the slight error in the camera pose estimated by SfM can cause discrepancies between the rendered image and the ground truth. Some other error pixels are located on small distant object parts, such as small leaves in the background, which are more difficult to reconstruct. The comparison of rendered images demonstrates that our variational multi-scale representation is capable of providing the extra functionality of uncertainty estimation while maintaining the performance of reconstructing various kinds of scenes with high fidelity.

## 4.4 Removing Noisy Floaters with Posterior Uncertainty

When reconstructing unbounded scenes such as those in Mip-NeRF 360, the training camera trajectories are around the foreground object. Therefore, insufficient information regarding the background is contained in the collected images, and the geometry of the distant background object is hardly reconstructed. As a result, the generalization ability over novel views on those regions is poor, and

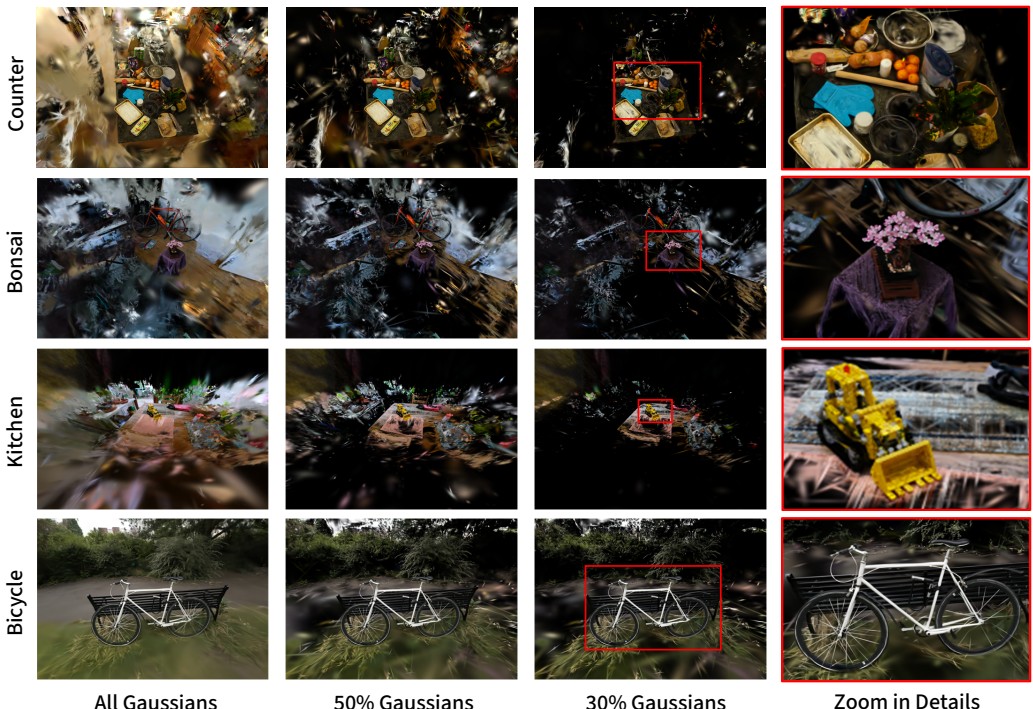

|             |             |             |                 |
| Counter     |             |             |                 |
| Bonsai      |             |             |                 |
| Kitchen     |             |             |                 |
| Bicycle     |             |             |                 |
| All Gaussians | 50% Gaussians | 30% Gaussians | Zoom in Details |

Figure 5: The results of noisy Gaussian removal on Mip-NeRF 360 scenes. By gradually deleting the Gaussians with large posterior uncertainty, our method removes the blurred floaters. The object of interest remains complete after the clean-up.

moving testing cameras away from the training trajectory leads to significant drops in visual quality. In order to obtain a high-fidelity radiance field, one can remove these noisy Gaussians manually with editing tools. However, this process requires a large amount of human labor.

Our uncertainty estimation can be used to automatically remove floaters by deleting Gaussians where their parameters have relatively large posterior uncertainty estimated. The results of floater removal are shown in Figure 5. We retain 50% and 30% of the Gaussians with smaller posterior uncertainty in their parameters. From a distant camera view, the novel view of the background scene fails to be synthesized due to the lack of the multi-view supervision signal. With the increasing number of removed Gaussians, the noisy Gaussians with large scale and irregular covariance in the background are removed gradually. After removing 70% of the Gaussians in the scene, the floaters in the background scene are mostly eliminated while leaving the clear Gaussians that capture the complete object of interest. By using our method, we can obtain clear foreground objects for further editing and presenting in VR/AR scenes.

## 5 Conclusion

In this paper, we have proposed a probabilistic framework to address the uncertainty estimation problem in the 3DGS algorithm. Different from previous work, we have identified the benefits of building multi-scale presentations to enhance the diversity of parameter space samples when performing Bayesian inference. Our method improves the efficiency of variational inference by parameter sharing and sampling only a subset of attributes of the model. Experimental results have demonstrated the accuracy of our method in uncertainty estimation and its effectiveness in removing floaters. The potential usages of our method include quality assessment of 3DGS scenes, robotics navigation and guided interactive active data acquisition.

# 6 Acknowledgement

This work was supported in part by the NSFC / Research Grants Council (RGC) Joint Research Scheme under the grant: N_HKBU214/21, and the RGC Senior Research Fellow Scheme under the grant: SRFS2324-2S02.

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

## A  Optimizing Variational Inference Objectives

In Section 3.3, we have leveraged the KL-divergence as the optimization goal of the multi-scale offset table with variational inference. However, the objective given by KL-divergence is intractable. Therefore, we have to further transform the objective as follows:

$$
\begin{aligned}
d_{KL}\left[q_\phi(\chi)\|p(\chi\mid\mathcal{D})\right] &:= \int_\chi q_\phi(\chi)\log\frac{q_\phi(\chi)}{p(\chi\mid\mathcal{D})} \\
&= \mathbb{E}_{q_\chi(\chi)}\left[\log q_\phi(\chi) - \log p(\chi\mid\mathcal{D})\right] \\
&= \mathbb{E}_{q_\phi(\chi)}\left[\log q_\phi(\chi)\right] - \mathbb{E}_{q_\phi(\chi)}[\log p(\chi,\mathcal{D}) - \log p(D)] \\
&= \mathbb{E}_{q_\phi(\chi)}\left[\log q_\phi(\chi) - \log p(\chi,\mathcal{D})\right] + \log p(\mathcal{D}),
\end{aligned}
$$

where $\phi$ is the parameter of variational distribution, which we stored in the offset. We can transform this objective to the Evidence Lower BOund (ELBO) by considering only the expectation part, and taking the expectation of the term over the data distribution $\mathcal{D}$. This yields a tractable optimization objective to maximize as follows:

$$
\begin{aligned}
\tilde{L}(\phi) &= \mathbb{E}_{\mathcal{D},q_\phi(\chi)}\left[\log p(\chi,\mathcal{D}) - \log q_\phi(\chi)\right] \\
&= \mathbb{E}_{\mathcal{D},q_\phi(\chi)}\left[\log p(\mathcal{D}|\chi) + \log p(\chi) - \log q_\phi(\chi)\right] \\
&= \mathbb{E}_{x,\hat{\mathbf{c}}}\left[\mathbb{E}_{\chi\sim q_\phi(\chi)}\left[\log p(\hat{\mathbf{c}}=\mathbf{c}\mid x,\chi) - d_{KL}(q(\chi)\|p(\chi))\right]\right],
\end{aligned}
$$

where $\log p(\hat{\mathbf{c}}=\mathbf{c}\mid x,\chi)$ is the log-likelihood of ground truth color, $p(\chi)$ is our multi-scale prior distribution, $q(\chi)$ is the variational distribution. In our method, we assume Laplace distribution for the predictive variable $\mathbf{c}$, then the loss function to optimize the prediction is the $\mathcal{L}_1$ loss function.

## B  Prior Distribution of Opacity Offset

To get the opacity offset $\chi_\alpha$, we first sample $\eta$ from the normal distribution, whose parameters are learned and stored in the offset table $\phi$. Then, we apply a mapping using a sigmoid function with temperature $\kappa$ to get the opacity offset $\chi_\alpha$:

$$
\chi_\alpha = \frac{1}{1+e^{-\kappa\cdot\eta}}. \tag{8}
$$

The offset $\chi_\alpha$ is applied to opacity $\alpha$ by multiplication: $\alpha^* = \alpha\cdot\chi_\alpha$. To derive the prior distribution for the offset $\chi_\alpha$, one can use the change of variable technique. Firstly, the inverse transformation from $\chi_\alpha$ to $\eta$ is:

$$
\eta = \frac{1}{\kappa}\ln\left(\frac{\chi_\alpha}{1-\chi_\alpha}\right). \tag{9}
$$

The Probability Density Function (PDF) of $\eta$ is given by the normal distribution:

$$
p(\eta) = \frac{1}{\sqrt{2\pi\sigma^2}}e^{-\frac{(x-\mu)^2}{2\sigma^2}}, \tag{10}
$$

where $\mu$ and $\sigma$ are the parameters of the normal distribution. Using the change of variables formula, we have:

$$
p(\chi_\alpha) = p(\eta)\left|\frac{d\eta}{d\chi_\alpha}\right| \tag{11}
$$

$$
= \frac{1}{\sqrt{2\pi\sigma^2}}e^{-\frac{\left(\frac{1}{\kappa}\ln\left(\frac{\chi_\alpha}{1-\chi_\alpha}\right)-\mu\right)^2}{2\sigma^2}}\left|\frac{1}{\kappa\cdot\chi_\alpha(1-\chi_\alpha)}\right|. \tag{12}
$$

The Cumulative Distribution Function (CDF) of opacity offset prior $\chi_\alpha$ is shown in Figure 6. The intuition for this offset prior is that we want to apply a small perturbation to the opacity in the variational inference. As shown in the CDF, with a positive $\mu$, the odds of offset $\chi_\alpha$ approaching 1 are high. Also, the sigmoid function is numerically stable in optimization. For computational simplicity, we minimize the KL divergence between the variational distribution and prior of $\eta$, instead of $\chi_\alpha$.

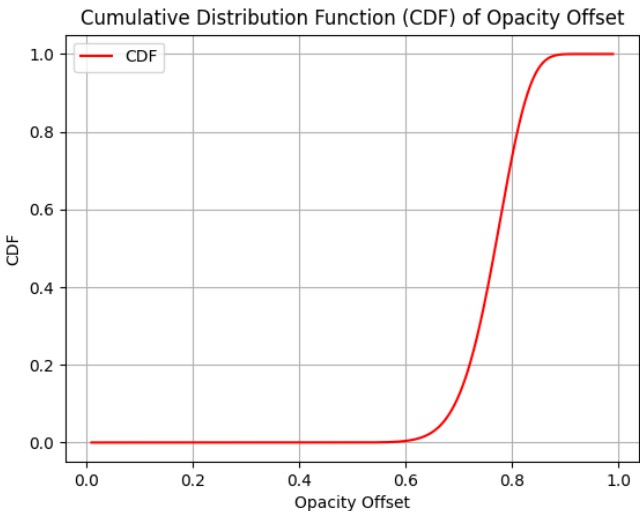

Figure 6: The Cumulative Distribution Function (CDF) of opacity offset prior.

## C   Rendering Time Analysis

Table 3: Inference time for variants of our method and the ensemble method.

|  | **Inference Time** (s) |
|---|---|
| Ensemble GS (x10) | $0.27 \pm 0.05$ |
| Ours$_{full}$ | $0.12 \pm 0.04$ |
| Ours$_{\mathbf{p},S,\mathbf{c}}$ | $0.08 \pm 0.02$ |
| Ours$_{\mathbf{p},S,\alpha}$ | $0.06 \pm 0.02$ |

The time for inferring a frame using our method compared with the ensemble method is shown in Table 3. We test on `torch` scene in the LF dataset with a single NVIDIA A100 GPU, and take the average time of $1,000$ frames. Ours$_{full}$ means offset with all attributes, Ours$_{\mathbf{p},S,\mathbf{c}}$ means offset with position, scaling and color. Ours$_{\mathbf{p},S,\alpha}$ means offset with position, scale and opacity, which is the setting of our main experiments. The running time shows that our design choice can achieve the lowest inference time.

## D   Active Learning Experiments

In active data acquisition of 3DGS, image collection and the 3DGS model training are performed alternately. Our goal is to maximize the model quality with the same number of images used. At each image collection step, the most informative image is selected via an acquisition function, in our case the uncertainty of the rendered image. By using our uncertainty estimation method as the acquisition function, we can indicate where the model is uncertain about and acquire more data around there.

We perform a simple experiment on active data acquisition of 3DGS on the LLFF dataset. Specifically, the original training dataset serves as the candidate image pool, and 10% of the images are randomly chosen for training initially. Then, one image is chosen for every 500 steps until in total 30% of images are used. We render our uncertainty map and aggregate the pixel values to choose the most

uncertain image from the pool as the next image added to the training set. After all images are chosen, the 3DGS model is further trained for 7K steps. The densification interval is 100 steps, and the spawning interval is 500 steps, and both operations are performed until training ends. As shown in Table 4, we found that the view synthesis quality of active 3DGS with our uncertainty estimation is better than choosing images randomly. As the experiment setting for active learning is intricate, we prefer to fully investigate the application of our uncertainty estimation on active learning, which is a limitation of this paper.

Table 4: The experiment on active learning with our uncertainty estimation.

|        | PSNR  | SSIM | LPIPS |
|--------|-------|------|-------|
| Random | 20.97 | 0.65 | 0.234 |
| Ours   | 21.35 | 0.69 | 0.212 |

## E   Ablation Study on the Number of Spawned Gaussian

We compare the view synthesis and uncertainty estimation performance using $K \in 1, 5, 10$ number of finer-level Gaussians spawned in the offset table. We train on all 8 scenes in the LLFF dataset and report the average results in Table 5. We found that improving the number of finer-level Gaussians $K$ shows a notable increase in the quality of uncertainty estimation. More finer level Gaussians improve the sample space diversity, therefore providing precise estimation of model parameter uncertainty and novel views.

Table 5: Ablation study on the number of spawned Gaussians.

|                                        | PSNR  | SSIM  | LPIPS | AUSE | NLL  |
|----------------------------------------|-------|-------|-------|------|------|
| 1  Finer Gaussisans                    | 23.42 | 0.792 | 0.183 | 0.37 | 0.24 |
| 5  Finer Gaussisans                    | 23.94 | 0.795 | 0.178 | 0.34 | 0.27 |
| 10  Finer Gaussisans (Default Setting) | 23.97 | 0.806 | 0.172 | 0.32 | 0.23 |

## F   Additional Visualization

Additional visualization results of view synthesis and uncertainty estimation of rendered images on scenes from the LLFF dataset using our method are provided in Figure 7. The images are down-scaled to half the original size.

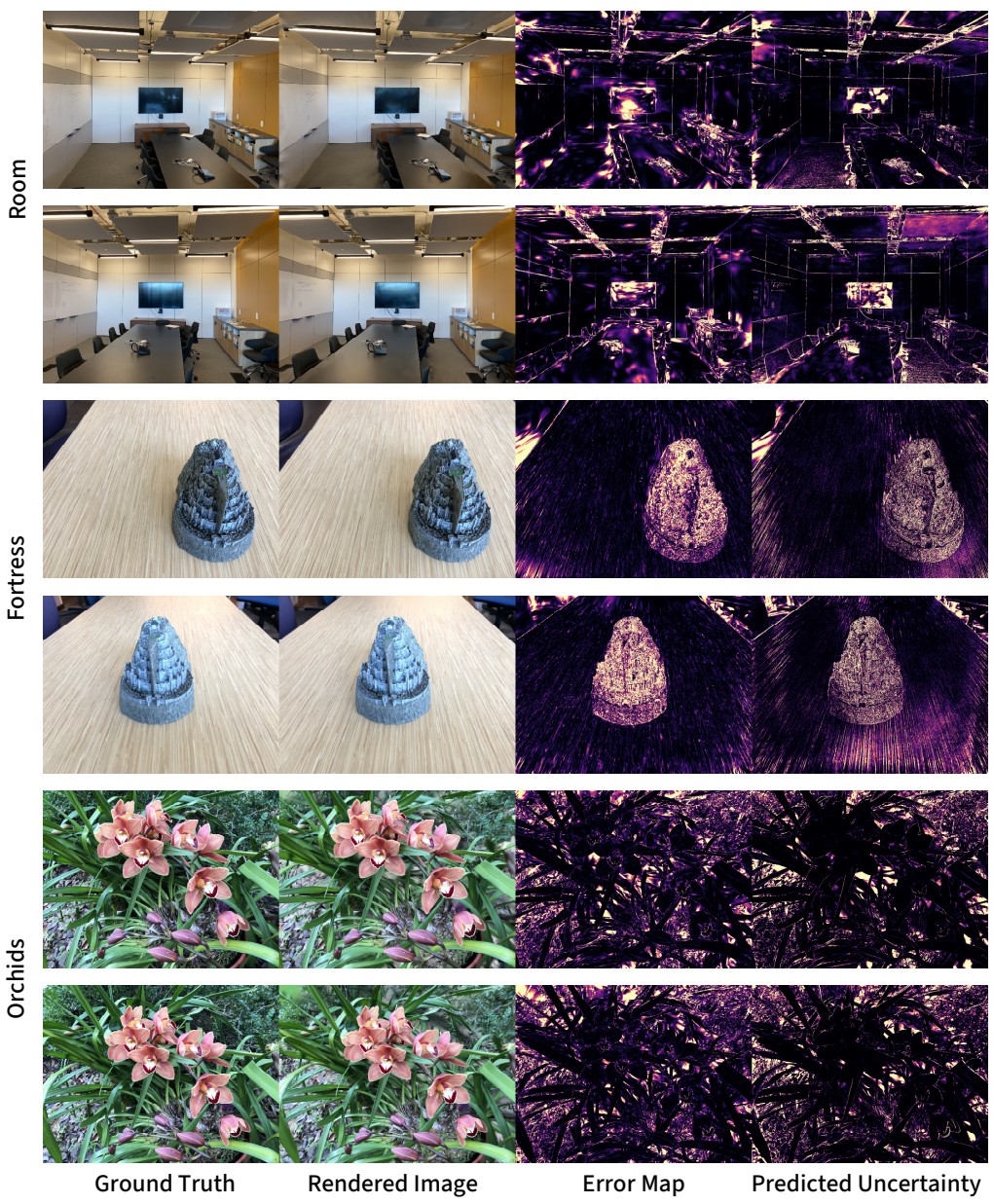

Figure 7: Addtional visualization results.

