# OpenReview forum: "Variational Multi-scale Representation for Estimating Uncertainty in 3D Gaussian Splatting"
_NeurIPS.cc/2024/Conference — NeurIPS 2024 poster_

### Official Review · Reviewer_ZTX9 · 2024-07-05

**Soundness:** 3
**Presentation:** 3
**Contribution:** 3
**Rating:** 7
**Confidence:** 4

**Summary:**

This work suggests a method to estimate uncertainty for the output of a 3D Gaussian Splatting (3DGS) model, by means of variational calculus.
Specifically, the authors split the 3DGS model to two hierarchical levels (or- two scales). The coarser "Base" level is very similar to vanilla 3DGS, but now each Gaussian contains a list of parameters $\{ \mu_i, \sigma_i\}$ that are used to spawn the "Finer" level Gaussians.
The latter "Finer" Gaussians are then fed into the standard 3DGS inference pipeline, only now these Finer Gaussians are fitted using the re-parametrization trick and the ELBO.
The experimental section compares to other uncertainty estimation methods and shows favorable results.

**Strengths:**

The method builds on the simplicity of 3DGS, and seems rather straightforward to implement. Given additional prior knowledge on ELBO based methods, the proposed method is rather easy understand from the text (with the small exception mentioned below).
The results show the power of this method compared to other methods (with the one exception mentioned below).

**Weaknesses:**

The method seems attractive, but needs to be better positioned w.r.t. previous works:
* FisherRF, (citation [42] in the paper) is mentioned in line 116 but is not compared against. The justification for the lack of numerical comparison is that "the posterior needs extensive computations", but the authors of [42] state they run on a modest card (RTX3090) in 70 FPS. At the very least, results from [42] can be easily added to Table 1 based on Table 4 in [42]. Additionally, the [code](https://github.com/JiangWenPL/FisherRF) of [42] was made public.
* latentSplat [arxiv 2403.16292](https://arxiv.org/abs/2403.16292) - while it operates under a **simpler setting than this work** , and while it is not yet accepted a peer-reviewed venue (AFAIK), I would recommend mentioning this method to prevent confusion.

**Questions:**

High level questions:
* Can the proposed method be applied directly on the **Base** level Gaussians? i.o.w. - why does one needs more than a single Gaussian in the Finer level?
* I would appreciate a deeper discussion on the similarities to [42].
---
While the method is rather straightforward, some design choices are not clear to me:
* why is $\mu_n$ in eq(6) is limited to this range?
* why is the spawning method of the Base level different from vanilla 3DGS?

**Limitations:**

The authors do not discuss limitation of their method.

---

> ### Author Rebuttal · Authors · 2024-08-06
>
> Thank you for the thoughtful review and the positive feedback! We address your comments and suggestions below.
>
> ## W1: Comparison with FisherRF
>
> We present the results of evaluating the quality of depth uncertainty maps using FisherRF. Note that our training view selection and implementation of AUSE in MAE metric follows the code of Bayes’ Ray [7]. The performance of other methods is provided in Table 1 in the main paper, and the average AUSE of our method is superior to FisherRF.
>
> |        | africa | basket | statue | torch | average |
> |--------|--------|--------|--------|-------|---------|
> | FisherRF | 0.29   | 0.27   | 0.31   | 0.42  | 0.32    |
> | Ours   | 0.18   | 0.17   | 0.13   | 0.29  | 0.19    |
>
> In our original submission, the choice of not presenting FisherRF results is mainly because all other methods we listed can compute not only depth uncertainty maps but also RGB uncertainty maps, which forms our complete evaluation. However, FisherRF only reports the depth uncertainty results in the paper. The provided code only contains depth uncertainty map rendering, where they compute the per Gaussian uncertainty and perform alpha blending as the depth uncertainty map.
>
> Regarding computational efficiency. We would to further clarify that FisherRF actually requires a pre-computation step to compute the Hessian for all the Gaussians. For the basket scene in the LF dataset, we use 29.7s in a single V100 card to compute these Hessian. Then, the uncertainty map can be rendered for around 70 FPS. Instead, our method does not require any pre-computation step for rendering RGB uncertainty.
>
> ## W2: Discussion about LatentSplat
>
> LatentSplat builds variational distribution for Gaussian features and achieves generalizable reconstruction of radiance field. Thank you for pointing this out, and we will discuss more and cite this work in the related work section.
>
> ## Q1: Why do we need more than one single Gaussian in the Finer level?
>
> Please refer to the results in Q2 in the global response.
>
> Actually, we found that the number of finer level Gaussian is rather important for the quality of uncertainty estimation. The purpose of multiple finer Gaussian is to encourage a variety of scale distribution among different finer level Gaussians that are attached to the same base Gaussian. This leads to more diversified samples in inferring the uncertainty using the learned posterior, which helps improve the accuracy of uncertainty.
>
> ## Q2: Deeper discussion on the similarity to FisherRF
>
> The similarity between our method and FisherRF is that they both can estimate the uncertainty explicitly for each point (Gaussian) in 3DGS. However, the parameter uncertainty in FisherRF $\mathbf{H}^{\prime \prime}\left[\mathbf{w} \mid D^{\text {train }}\right]$ is approximated with the diagonal elements of the Hessian matrix of the parameter $\mathbf{w}$ given the training data $D^{\text {train}}$.
>
> This approximation is made by employing assumptions like the predictive distribution should be a normal distribution with the mean equal to the maximum a posteriori solution and precision equal to the Fisher Information (Equation 18 in [6]). Instead, we follow a Variational Bayesian approach, in which we set priors for parameters, then minimize the KL divergence between prior and variational distribution to infer the variance of model parameters as the uncertainty.
>
> ## Q3: Why $\mu_{n}$ is limited to this range
>
> The prior distribution of offset is set as this uniform distribution so that the scale of $K$ number of finer level Gaussian after offset would become $S_{offset} = S_{base}+\mu_n \sim U (S_{base}/K, S_{base})$, which encourage that the finer Gaussians scale not to exceed $S_{base}$, the scale of base Gaussian they attached or become too small and lose functionality. This setting enables the multi-scale representation by learning the distribution of scale of finer level Gaussian.
>
> ## Q4: Why is the spawning method of base level different from vanilla 3DGS
>
> We would like to clarify that densification and spawning are different operations.
> Spawning refers to the operations that select and offset base level Gaussians to construct finer level Gaussian. The densification of Gaussian is to clone all the attributes of Gaussians and translate the new ones.
>
> For example, in our experiment, we train the LLFF scenes for 12K steps in total, during which the densification is performed for every 500 steps and spawning is performed for every 4K steps.
>
> Note that both base Gaussian and non-base Gaussian can perform densification, the difference is that the densification of base Gaussian also clones the attached finer Gaussians. Apart from that, the implementation of densification is identical to the original 3DGS. Thank you for pointing this out, and we will add the above details in the revision.
>
> If you have further questions please feel free to raise them. We would be glad to provide materials and discuss!

---

> > ### Comment · Reviewer_ZTX9 · 2024-08-09
> > **Thanks for the deatiled reply**
> >
> > The authors have thoroughly addressed all of my questions and concerns,
> > and I stand by my positive score.
> >
> > ---
> > Indeed, I mistakenly asked about spawning rather than densification.
> > If I understand correctly - you create/duplicate new base-Gaussians every 500 iterations, and re-spawn new finer-Gaussians every 4K iterations.

---

> > > ### Author Response · Authors · 2024-08-09
> > >
> > > We sincerely appreciate the feedback from the reviewer! The difference between densification and spawning operations is exactly what you describe above.
> > >
> > > Thank you for your positive score on our work!

---

### Official Review · Reviewer_XD3Y · 2024-07-05

**Soundness:** 4
**Presentation:** 3
**Contribution:** 4
**Rating:** 7
**Confidence:** 5

**Summary:**

This paper proposes an uncertainty estimation method for the 3D Gaussian Splatting (3DGS) radiance field reconstruction algorithm. The proposed methods leverage the multi-scale properties that lie inherently in a vast amount of Gaussian ellipsoids to improve the performance of variational inference. The paper validates quantitatively that the proposed method aligns with the novel view synthesis error better than previous methods on NeRF uncertainty estimation and naïve methods on 3DGS uncertainty estimation. The experiments also validate an interesting application of the proposed uncertainty estimation methods: to determine noisy Gaussians and remove them thus reduce the floater artifacts.

**Strengths:**

1. The motivation of this paper is clear. The problem of increasing diversity in the sampling of variational inference is challenging, and the idea of developing a multi-scale representation in 3DGS to solve it is intriguing.

2. The method is sound and easy to follow. Representing the same scene with multiple scale levels is feasible in previous CG techniques. The proposed technique of learning the scene with different scales by assuming different variational distributions, and estimating the predictive uncertainty by sampling from these “multi-scale posteriors” is clear. Designing an offset table to increase inference efficiency is a plus.


3. The experiments demonstrate the effectiveness of the method in uncertainty estimation by comparing AUSE and NLL. The synthesized image quality in Table 6 is better than NeRF based method, and the noisy Gaussians removal experiments are interesting.

**Weaknesses:**

1. Ambiguous details: i): In line 187, there are K alternative values in the offset table. If so, how many values that the offset table contain for one spawned Gaussian? ii): In Figure 3 (c), how are sampling and inference done from multiple spawned Gaussians? The Figure should contain more details to clarify the difference with other methods.

2. More ablation studies. i): Replacing the multi-scale prior in equation 6 with the same prior distribution for all layers can be compared to study the performance gain of the multi-scale method. ii): The impact of the quantity of spawned Gaussians in the offset table (line 174) on the reconstruction performance can also be studied.

**Questions:**

Please see weakness.

**Limitations:**

More comparing results with simpler methods such as naïve variational inference can be provided.

---

> ### Author Rebuttal · Authors · 2024-08-06
>
> Thank you for your acknowledgment of our work! We provide illustrations for your concerns below.
>
> ## W1: Ambiguous values in the offset table for each spawned Gaussian
>
> Since we only offset the position and scale in the offset table, and each offset table contains $K$ spawned Gaussians, then for each spawned Gaussians the length of the offset vector is 12 (6 for position offset distributions and 6 for scale), so the number of alternative values in the offset table is $12 \cdot K$.
>
> ## W2: Ambiguous Figure
>
> Generally, the inference pipeline of our method is to first sample from the spawned finer Gaussians and then sample from their learned posterior distribution. We will update the details in the figure.
>
> ## W3: How about setting the same prior for all finer levels
>
> Thank you for the question. Actually, if setting the same prior for all finer levels, the learned posterior would lean to the same distribution after training. Therefore, this method could be regarded as equivalent to using a single finer level Gaussian. We perform additional experiments to explore the effectiveness of this method, and provide the results on the LLFF dataset in Q2 in the global response. We found that using only one finer level Gaussian would largely decrease the uncertainty estimation performance.
>
> ## W4: The impact of the number of spawned Gaussians
>
> Generally, the performance would increase when there is more number of fine level Gaussians. This is because more spawned Gaussians provide more capacity to fit the posterior distribution. We provide the comparison results and analysis of using 1, 5 and 10 spawned Gaussians in Q2 of the global response to validate this.
>
> ## L1: Comparison with naive variational inference
> Please also see the answer in W3.
>
> We’ll be happy to address any further questions, please feel free to raise them.

---

> > ### Comment · Reviewer_XD3Y · 2024-08-13
> >
> > The rebuttal from the author has solved my concerns. I will keep my rating as accept. The author should address the problems raised in the comments in the revised version.

---

> > > ### Author Response · Authors · 2024-08-13
> > >
> > > Thank you for the feedback!
> > >
> > > We will improve the presentation quality, and add the above ablations in the revision.

---

> ### Author Response · Authors · 2024-08-12
>
> Dear Reviewer,
>
> We appreciate your precious time and efforts in reviewing our work! We look forward to addressing any concerns you may have during the remainder of the discussion period.
>
> Best Regards,
> Authors

---

### Official Review · Reviewer_4RNR · 2024-07-10

**Soundness:** 2
**Presentation:** 2
**Contribution:** 2
**Rating:** 4
**Confidence:** 4

**Summary:**

The paper aims to quantify uncertainty in the learning pipeline in 3DGS. To this end, the author(s) proposed to leverage explicit scale information to build variational multiscale 3D gaussians leading to the construction of diversified parameter space samples. This results in the proposition of a multiscale variational inference framework for uncertainty estimation in the 3D Gaussian model. Experimental results on popular benchmark datasets are shown to demonstrate the efficacy of the proposed method.

**Strengths:**

1. The paper solves an extremely useful problem for robot vision and control applications: quantifying uncertainty in 3D Gaussian splatting. Uncertainty modeling is essential for developing robot vision-based automation systems that are robust, safe, efficient, and capable of operating in complex and dynamic environments. By explicitly accounting for uncertainties, this approach can help design more effective and reliable robotic vision systems. Therefore, this paper clearly attempts to address a significant problem.

2. Uncertainty quantification on explicit 3DGS by exploiting both the model space diversity and efficiency. Such a modeling strategy ensures efficient and compact model output.

**Weaknesses:**

1. The results are not as impressive as mentioned in the abstract and in the introduction of the paper, i.e., state-of-the-art.

2. The approach to use only scale can cause problems with points that are far from the camera. This is critical particularly for the formulation that takes both scale and rotations to model the covariance —refer to 3DS original work. Kindly comment.

3. Writing of the paper can be improved.

        a. can not -> cannot

        b. robotics navigation -> robot navigation

        c. less samples of parameters as possible -> parameters samples as few (less) as  possible.

        d. spawn strategy -> not clearly explained while introducing this term for the first time in the paper, i.e.,  what author(s) mean by this term—referring to contribution. Explaining at least briefly here will improve the draft.

        e. Evaluation is generally not considered as contribution (contribution 3)

        f. Section 2.2 line 97 “Rearly…this task”. Kindly rephrase this line, it is confusing in the current form.

        g. Line 228: “the original is”...?

        h. Line 234: Then-> then

        There are more typos… kindly correct them.

        i. Missing references in implicit uncertainty modeling Lee et al. RAL 2022 “uncertainty guided policy for active robotic 3d reconstruction using neural radiance fields”

**Questions:**

1. Kindly clarify how removing uncertain 3D Gaussian still leads to complete scene images visualization on screen (rasterization).

2. Kindly provide details on how distance of the points from the camera affects the uncertainty quantification.

3. It is well-known that COLMAP 3D and camera pose are not perfect. How structure from motion uncertainty contributes to the current approach. Kindly comment.

**Limitations:**

Although a few limitations are obvious from the experiments section. It is not explicitly mentioned in the paper.

---

> ### Author Rebuttal · Authors · 2024-08-06
>
> Thank you for the kind and helpful comments! We will address your concerns below.
>
> ## W1: The results are not as impressive as claimed
>
> Firstly, we’d like to clarify that the ensemble method is a naive baseline that trains 10 vanilla 3DGS with different random seeding, which incurs an extreme computational burden compared to other methods. Therefore, the ensemble method serves as a performance upper bound, with the same practice in [7].
>
> Apart from the ensemble method, our method is optimal in most cases. The only exception is the AUSE and NLL metric on the LF dataset, where our method is inferior to CF-NeRF, in Table 2. We discuss this result on line 250-252. On the other hand, in this setting, our view synthesis quality is better than other methods, which is another primary goal of building uncertainty-aware 3DGS models.
>
> ## W2: The approach to use only scale can cause problems with points that are far from the camera; scale and rotation together form the covariance
>
> We provide qualitative results regarding the points far from the training camera in Figure 2 in the attachment. In unbounded scenes from MipNeRF 360 dataset, using all training images, our method successfully renders both uncertainty maps and novel views of the distance background. Technically, although we refer to our method as multi-scale representations, as shown in Equation 6, we offset not only scale but also position to spawn the finer Gaussian.
>
> Additionally, the scale of the finer Gaussian is restricted by our setting of prior distributions. The learned posterior of $K$ finer Gaussian scale would approach $U (S_{base}/K, S_{base})$, which means that the scale of finer Gaussians would be constrained softly to avoid unlimited variation in scale compared to the base Gaussian they attached. Therefore, for base Gaussians far from the camera, the scale of their spawned finer Gaussian is still adaptive, without harming the rendering quality. In practice, we found that the distributions of rotation over the scene are rather random and irregular compared to the scale, while the scale varies with object size. Therefore, we chose to decompose the covariance and only offset the scale to form our multi-scale representation.
>
> ## W3: Writing and reference problems
>
> Thanks for your careful inspection! We will fix the raised problems and improve the writing thoroughly in the revision.
>
> The RAL paper proposes to use the entropy along the ray weight to evaluate pixel-wise uncertainty, which is used to guide active reconstruction. We will discuss and cite the RAL paper in the related work section.
>
> ## Q1: How does removing uncertain 3D Gaussian still lead to complete scene renderings
>
> Please refer to Q1 in the global response, where we visualize that when removing 10% of the Gaussians, the complete scene is preserved and small floaters are cleaned. We also illustrate why in the original paper Figure 1 and Figure 4 we remove at least 50% of the noisy Gaussians.
>
> ## Q2: Details on how the distance of the points from the camera affects the uncertainty
>
> When estimating the per point (Gaussian) uncertainty in the MipNeRF 360 dataset, usually the model is more uncertain about the points (Gaussians) distant from the training camera trajectory. Generally, the uncertainty value of points depends on how well they are covered by multiple views (cameras) without occlusion. The distant, background points have less content provided from multiple views in a casually captured dataset, which makes the background modeling difficult.
>
> When rendering the uncertainty map, the points (Gaussians) far from testing cameras are projected to a smaller pixel region in image space, due to the inherent properties of perspective projection. However, the intensity of uncertainty pixels is consistent within the projected area in image space.
>
> ## Q3: How does COLMAP uncertainty contribute to the current approach
>
> In 3DGS training, the COLMAP generates sparse point clouds and camera poses as the input of 3DGS. Thus, from the 3DGS point of view, the uncertainty contained in camera poses generated by the structure from motion can be categorized as aleatoric uncertainty, which inherently exists in the input data to 3DGS [5]. The aleatoric uncertainty, together with epistemic uncertainty that exists in the model parameters of 3DGS, results in the predictive uncertainty estimated in our method.
>
> ## L1: More discussion on limitations
>
> For quantitative results, we discuss the limited results in line 250-252 of the original paper. We will talk more about the qualitative performance of rendering uncertainty maps in MipNeRF 360 in the revision.
>
> Thank you again for your effort in reviewing our work! We’ll be glad to discuss if there are any further concerns.

---

> ### Author Response · Authors · 2024-08-12
>
> Dear Reviewer,
>
> We appreciate your precious time and efforts in reviewing our work! We look forward to addressing any concerns you may have during the remainder of the discussion period.
>
> Best Regards,
> Authors

---

### Official Review · Reviewer_UQFr · 2024-07-10

**Soundness:** 3
**Presentation:** 3
**Contribution:** 3
**Rating:** 6
**Confidence:** 4

**Summary:**

This paper proposed a novel multi-scale variational representation for 3D gaussian splatting to estimates its uncertainty. Specifically, this paper introduced a spawn strategy to split a base gaussian into multiple sub gaussians and apply variational inference to estimate the uncertainty.

**Strengths:**

(1), Estimating uncertainty in the 3D reconstruction / novel view synthesis systems is a critical task and has great application potential.

(2), The paper proposed a novel representation and solid mathematical derivation to solve this problem.

(3), The experiment shows strong relevance between the estimated uncertainty and the rendering error.

**Weaknesses:**

(1), Authors claimed one of the potential application is guided interactive active data acquisition. It would be great to have more detailed illustration or even experimental exploration.

**Questions:**

(1), For the floater removal experiment, is removing the background actually the desired effect for real world use case? In my understanding , a good uncertainty-based gaussian pruning approach should be able to remove the blurry gaussians to improve the rendering clarity while preserving most of the objects regardless its foreground or background.

(2), For the related works section, there are some additional related uncertainty quantification works could be discussed:
[1] Naruto: Neural active reconstruction from uncertain target observations. CVPR 2024
[2] Active neural mapping. ICCV 2023

**Limitations:**

The uncertainty qualification requires (additional) sub gaussian construction, might potentially increase the computational burden.

---

> ### Author Rebuttal · Authors · 2024-08-06
>
> Thank you for your insightful comment and positive feedback! We address your concerns as follows.
>
> ## W1: Discuss more about active data acquisition.
>
> In active data acquisition of 3DGS, image collection and the 3DGS model training are performed alternately. At each image collection step, the most informative image is selected via an acquisition function to maximize the model quality with the same number of images used. Our uncertainty estimation method can contribute to this acquisition function, indicating where the model is uncertain about and acquiring more data around there.
>
> We perform a simple experiment on active data acquisition of 3DGS on the LLFF dataset. Specifically, the original training dataset serves as the candidate image pool, and 10% of images are randomly chosen for training initially. Then, one image is chosen for every 500 steps until 30% of images are used. We render our uncertainty map and aggregate the pixel values to choose the most uncertain image from the pool as the next image added to the training set. After all images are chosen, the 3DGS model is further trained for 3K steps. The densification interval is 100 steps, and the spawning interval is 500 steps, and both operations are performed until training ends.
>
> As shown in the Table below, we found that the view synthesis quality of active 3DGS with our uncertainty estimation is better than choosing images randomly. Due to the various detailed settings in the active data acquisition task, we prefer to fully evaluate the performance of our uncertainty in active learning in the revision of this paper.
>
>
> |        | PSNR  | SSIM | LPIPS |
> |--------|-------|------|-------|
> | Random | 20.97 | 0.65 | 0.234 |
> | Ours   | 21.35 | 0.69 | 0.212 |
>
>
> ## Q1: Floater removal should clear blurry Gaussians regardless of foreground or background.
>
> Please refer to Q1 in global response, where we show that our method can clean foreground noisy Gaussians.
>
> ## Q2: Discussion on related work.
>
> Thank you for pointing these out! Different from our uncertainty estimation method, Naruto [8] learns the uncertainty of depth by the Negative Log Likelihood loss function. Active neural mapping [9] leverages the artificial neural variability [1] to indicate the predictive uncertainty. We will compare and cite them in the related work section.
>
> ## L1: Constructing sub Gaussians might increase the computational burden
>
> We analyze specifically the extra computational cost from the following perspective:
>
> **Rendering RGB image**: The image rendering speed is basically the same as vanilla 3DGS rendering, since we do not require multiple sampling from variational distribution.
>
> **Calculating Per Gaussian Uncertainty for Pruning**: Calculating per Gaussian uncertainty for pruning simply aggregates the variance of learned posteriors for each Gaussian, which takes less than 1s for the garden scene in MipNeRF 360 in a V100 GPU.
>
> **Rendering Uncertainty Map**: Rendering the uncertainty map requires sampling from the variational distribution. If taking $N$ samples, the rendering cost would be $N$ times the vanilla rendering cost. In our experiments, we set $N$ equal to 10, the same as the number of finer gaussian. This cost growth follows the common practice in Variational Bayes approaches such as CF-NeRF [4] and others [2], [3].
>
> **Memory**: The extra memory cost for the offset table of a base Gaussian is $12 \cdot (K-1)$ floating numbers while $K$ is the number of finer Gaussians. We only build offset tables for base Gaussians which takes up around 50% of all Gaussians in the original scene. If rendering uncertainty maps is not required, we can retain only one entry in the offset table to render RGB images after pruning the noisy Gaussians.
>
> We’ll be glad to discuss if you have any further concerns.

---

> > ### Comment · Reviewer_UQFr · 2024-08-08
> > **Thank you for the response**
> >
> > I appreciate the author for the detailed response. This rebuttal resolves most of my concern, I would keep my rating as weak accept. Authors are encouraged to add the active data acquisition part to the main paper or at least the supplementary materials.

---

> > > ### Author Response · Authors · 2024-08-09
> > >
> > > Thank you for affirming our response! We will add more detailed content on active data acquisition to the revision. We welcome discussion on any further questions.

---

### Author Rebuttal · Authors · 2024-08-06

We would like to thank all the reviewers for their time and efforts in reviewing our paper and providing insightful comments. Here we address two common questions from the reviewers in this global response, and the individual questions from each reviewer are provided separately.

## Q1: To Reviewer [UQFr, 4RNR]: Why does the floater removal experiment remove the background? Can we retain the background?

Firstly, we observe that when training the 3DGS on MipNeRF 360 unbounded scenes, the majority of the floaters lie in the background region. This is because the training camera trajectory is placed around the centering object while providing insufficient multi-view information to reconstruct the background. Thus, we visualize the floater cleaning performance by removing at least 50% of the Gaussians to validate that our method can remove most of these noisy floaters. Moreover, our visualization results are consistent with Figure 1 in Bayes Ray [7], which also removes the noisy background of an unbounded scene represented by NeRF.

We further show that our method can also perform the floater removal effect for noisy Gaussian in the foreground. As shown in Figure 1 in the attachment, when removing only 10% of the Gaussian and visualizing from a close view, our method can also remove smaller floaters in the foreground to improve the clarity of the synthesized view, while keeping the background complete.

## Q2: To Reviewer [XD3Y, ZTX9]: Comparison with simpler methods, such as using less or even one finer level Gaussian.

We compare the view synthesis and uncertainty estimation performance using $K \in {1, 5, 10}$ number of finer level Gaussians spawned in the offset table. Same as Section 4.2 in the original paper, we train on all 8 scenes in the LLFF dataset and report the average results. We found that improving the number finer level Gaussian $K$ shows a notable increase in the quality of uncertainty estimation. More finer level Gaussians improve the sample space diversity, therefore providing precise estimation of model parameter uncertainty. Nevertheless, the quality of novel views fluctuates when $K$ changes. We think this is because 5 spawned Gaussians are enough to represent the scene while the quality of uncertainty could be further promoted by more spawned Gaussians.

|                                    |       |       |       |      |      |
|------------------------------------|-------|-------|-------|------|------|
|                                    | PSNR  | SSIM  | LPIPS | AUSE | NLL  |
| 1 Finer Gaussisans                      | 23.67 | 0.791 | 0.194 | 0.52 | 0.47 |
| 5 Finer Gaussisans                     | 23.92 | 0.797 | 0.179 | 0.43 | 0.34 |
| 10 Finer Gaussisans  (Default Setting)  | 23.84 | 0.805 | 0.186 | 0.38 | 0.32 |

We thank the reviewers again for their valuable feedback, and sincerely look forward to discussing any further questions!

## Reference

[1] Xie, Zeke, et al. "Artificial neural variability for deep learning: On overfitting, noise memorization, and catastrophic forgetting." Neural computation (2021).

[2] Kingma, Durk P., Tim Salimans, and Max Welling. "Variational dropout and the local reparameterization trick." NeurIPS (2015).

[3] Blundell, Charles, et al. "Weight uncertainty in neural network." International conference on machine learning. ICML (2015).

[4] Shen, Jianxiong, et al. "Conditional-flow nerf: Accurate 3d modelling with reliable uncertainty quantification." ECCV (2022).

[5] Kendall, Alex, and Yarin Gal. "What uncertainties do we need in bayesian deep learning for computer vision?." NeurIPS (2017).

[6] Kirsch, Andreas, et al. “Unifying Approaches in Active Learning and Active Sampling via Fisher Information and Information-Theoretic Quantities.” TMLR (2022).

[7] Goli, Lily, et al. "Bayes' Rays: Uncertainty Quantification for Neural Radiance Fields." CVPR (2024).

[8] Feng, Ziyue, et al. "Naruto: Neural active reconstruction from uncertain target observations." CVPR (2024).

[9] Yan, Zike, et al. "Active Neural Mapping." ICCV (2023).

---

> ### Comment · Reviewer_ZTX9 · 2024-08-09
> **Thanks for the analysis**
>
> What happes for 15 or even 20 finer Gaussians? (I looked for such a table in the paper but didn't find)

---

> > ### Author Response · Authors · 2024-08-09
> >
> > Thank you for the question!
> >
> > In our experiments, we found that spawning for more than 10 Gaussians shows a non-significant gain in uncertainty estimation metric. Actually, spawning between 5 and 10 already show a tradeoff between the quality of image and uncertainty, while 10 finer Gaussians lead to better uncertainty quality. Therefore, we chose 10 as the default setting and did not present the result of more Gaussians considering the extra computational cost. However, we only tested on LLFF scenes and the performance on more benchmarks might vary. We will add a more detailed ablation study in the revision on the effect of the spawned gaussian number, as raised by the reviewer [XD3Y, ZTX9].

---

### Decision · Program_Chairs · 2024-09-25

**Decision:**

Accept (poster)

**Comment:**

This paper was reviewed by four experts in the field.  Based on the reviewers' feedback as well as the author feedback, the decision is to recommend the paper for acceptance.  The reviewers did raise some valuable comments that should be addressed in the final camera-ready version of the paper. The authors are encouraged to make the necessary changes to the best of their ability.    We congratulate the authors on the acceptance of their paper!